# Mixed-Field Source Localization Based on the Non-Hermitian Matrix

**Minggang Mo \* and Zhaowei Sun**

School of Astronautics, Harbin Institute of Technology, Harbin 150001, China; sunzhaowei@hit.edu.cn
\* Correspondence: 14B318027@hit.edu.cn; Tel.: +86-134-3900-4103

**Abstract:** In this paper, an efficient high-order multiple signal classification (MUSIC)-like method is proposed for mixed-field source localization. Firstly, a non-Hermitian matrix is designed based on a high-order cumulant. One of the steering matrices, that is related only with the directions of arrival (DOA), is proved to be orthogonal with the eigenvectors corresponding to the zero eigenvalues. The other steering matrix that contains the information of both the DOA and range is proved to span the same column subspace with the eigenvectors corresponding to the non-zero eigenvalues. By applying the Gram–Schmidt orthogonalization, the range estimation can be achieved one by one after substituting each estimated DOA. The analysis shows that the computational complexity of the proposed method is lower than other methods, and the effectiveness of the proposed method is shown with some simulation results.

**Keywords:** MUSIC; mixed-field; high-order; cumulant

## 1. Introduction

Source localization has been an important research topic for array signal processing [1–3]. This research topic is widely applied in many fields such as sonar and electronic surveillance [4], where the signals are often non-stationary [5]. The wavefront of a far-field source signal can be assumed to be a plane when it impinges on the receiver array. Each source can be localized with its corresponding direction of arrival (DOA). When the sources are located near the array, the wavefront of the impinging signal is spherical [6,7]. These near-field sources require both the DOAs and ranges to specify their location [8].

Plenty of researchers all over the world have been making an effort to contribute to the research of mixed-field source localization, and there are already many achievements. For far-field source localization, there is the multiple signal classification (MUSIC) in [9], estimating signal parameters via rotational invariance techniques (ESPRIT) in [10], and root-MUSIC in [11]. For near-field, many people directly extend the DOA estimation methods to estimate the DOA and range at the same time. For example, there are the two-dimension (2D) MUSIC method [12] and the 2D ESPRIT algorithm [13].

But in practice, the situation is very often that far-field and near-field sources exist simultaneously [14,15]. In recent years, many scholars have been concentrating on the research about the mixed-field source localization, and proposed many modified methods that can avoid the 2D search with a high computational complexity. A modified 2D MUSIC method was proposed to construct several cumulant matrices and the mixed-field sources are localized with several 1D searches [16]. Later, a mixed-order statistics MUSIC (MOS) was proposed in [17] that can reduce the computational complexity of [16]. However, all these existing methods require constructing two different Hermitian matrices and applying the eigenvalue decomposition (EVD) twice. In [18], a simplified 2D MUSIC method was proposed to localize near-field sources with constructing only one matrix. But it can only be applied for near-field source localization.

Therefore, we propose in this paper a simplified method to localize mixed-field sources that requires only one non-Hermitian matrix and one EVD while avoiding the 2D search. Firstly, with the high degrees of freedom, the high-order cumulant is adopted to design a non-Hermitian matrix, which can be expressed as the product of two different steering matrices. Then one steering matrix can be proved to be orthogonal with the eigenvectors corresponding to the zero eigenvalues, and the DOA of all the sources can be estimated with these eigenvectors like other methods. The other one can be proved to be related with the eigenvectors corresponding to the non-zero eigenvalues. At last, the ranges of the near-field sources can be estimated with these eigenvectors by applying the Gram–Schmidt orthogonalization.

The rest of this paper is organized as follows. The signal model is displayed in Section 2. Section 3 makes a detailed illustration of the proposed method, as well as the complexity analysis, to show the corresponding improvement. Several simulations are carried out in Section 4. At last, the whole paper is concluded in Section 5.

In this paper, $T$ means the transpose operation, $H$ is the conjugate transpose, and $*$ the complex conjugate. A bold capital letter symbolizes a matrix, and a bold letter in lower case stands for a vector, such as A and a respectively.

## 2. Signal Model

In this paper, the situation is considered where $K$ mixed-field narrow-band signals impinge on a uniform linear array (ULA). The ULA consists of $2M + 1$ elements as shown in Figure 1, with the inter-element spacing being $d$. The sources include $K_1$ far-field ones and $K_2$ near-field ones ($K_1 + K_2 = K$). The output with $T$ snapshots of the $m$th ($m \in [-M, M]$) sensor can be expressed as

$$y_m(t) = \sum_{k=1}^{K} s_k(t)e^{j\varphi_{mk}} + n_m(t), t = 1, 2, \ldots, T, \tag{1}$$

where $s_k(t)$ stands for the signal from the $k$th source, $n_m(t)$ represents the additive Gaussian noise (colored or white) at the $m$th sensor. Generally, the phase difference between the 0th and $m$th elements $\varphi_{mk}$ can be expressed as [19]

$$\begin{aligned}
\varphi_{mk} &= \frac{2\pi}{\lambda}(\sqrt{r_k^2 + (md)^2 - 2r_k md \sin \theta_k} - r_k) \\
&\approx \omega_k m + \phi_k m^2,
\end{aligned} \tag{2}$$

where

$$\omega_k = -\frac{2\pi d}{\lambda} \sin \theta_k \tag{3}$$

$$\phi_k = \frac{\pi d^2}{\lambda r_k} \cos^2 \theta_k \tag{4}$$

$\lambda$ is the wavelength of the source signal, satisfying $\lambda \geq 4d$, $r_k$ is the range of the $k$th source, and $\theta_k$ is the corresponding DOA. In the near-field situation, the sources are located in the Fresnel region ($r_k \in [0.62(R^3/\lambda)^{0.5}, 2R^2/\lambda]$, where $R = 2Md$ is the aperture of the ULA). However, for far-field sources, the ranges are beyond the Fresnel region and can be considered infinite. We have for far-field sources

$$\phi_k \approx 0. \tag{5}$$

Therefore, only the DOA is required for far-field source localization while both the DOA and range are necessary to localize the near-field source.

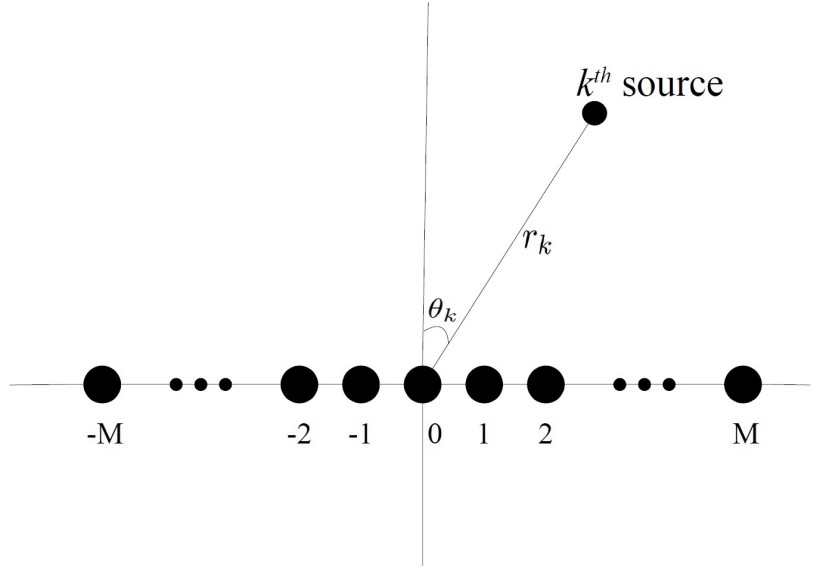

**Figure 1.** Near-field source localization with a uniform linear array (ULA).

The received signal can also be expressed in the matrix form as follows:

$$\mathbf{y}(t) = \mathbf{A}(\boldsymbol{\theta}, \mathbf{r})\mathbf{s}(t) + \mathbf{n}(t) \tag{6}$$

$\mathbf{y}(t)$ is the $(2M+1) \times 1$ received signal vector:

$$\mathbf{y}(t) = [y_{-M}(t), y_{-M+1}(t), \ldots, y_M(t)]^T, \tag{7}$$

$\mathbf{s}(t)$ is the $K \times 1$ signal vector from the $K$ sources:

$$\mathbf{s}(t) = [s_1(t), s_2(t), \ldots, s_K(t)]^T. \tag{8}$$

$\mathbf{A}(\boldsymbol{\theta}, \mathbf{r})$ is the steering matrix:

$$\mathbf{A}(\boldsymbol{\theta}, \mathbf{r}) = [\mathbf{a}(\theta_1, r_1), \mathbf{a}(\theta_2, r_2), \ldots, \mathbf{a}(\theta_K, r_K)]. \tag{9}$$

$\mathbf{a}(\theta_k, r_k)$ is the $(2M+1) \times 1$ steering vector:

$$\mathbf{a}(\theta_k, r_k) = [e^{j[(-M)\omega_k + (-M)^2\phi_k]}, \ldots,$$
$$e^{j(M\omega_k + M^2\phi_k)}]^T. \tag{10}$$

And $\mathbf{n}(t)$ is the $(2M+1) \times 1$ noise vector:

$$\mathbf{n}(t) = [n_{-M}(t), n_{-M+1}(t), \ldots, n_M(t)]^T. \tag{11}$$

Without loss of generality, the following assumptions (like in [14–19]) are made to guarantee the uniqueness of the localization:

(1)  The kurtosis of the signal is non-zero.
(2)  The DOAs of all sources are different.
(3)  The source signals are independent of each other as well as of all the noise.
(4)  The number of the sensors is greater than that of the sources.

## 3. Proposed Scheme

### 3.1. DOA Estimation for Mixed-Field Sources

High-order cumulant can resist the Gaussian noise effectively, no matter whether it is white or colored. When the statistic order is no smaller than 3, the cumulant of this kind of noise should be zero [20]:

$$cum\{n_m(t), n_n^*(t), n_p(t), \ldots\} = 0, \tag{12}$$

where $n_m(t)$, $n_n(t)$ and $n_p(t)$ are the noise received at the *m*th, *n*th and *p*th receiver respectively. The fourth-order cumulant is adopted in this paper, which is free from the Gaussian noise. Under Assumption 3 that the source signals are independent of all the noise, we have

$$
\begin{aligned}
& cum\{y_m(t), y_n^*(t), y_p(t), \ldots\} \\
= \ & cum\{\sum_{k=1}^{K} s_k(t)e^{j\varphi_{mk}} + n_m(t), \sum_{k=1}^{K} s_k(t)e^{j\varphi_{nk}} + n_n(t), \sum_{k=1}^{K} s_k(t)e^{j\varphi_{pk}} + n_p(t), \ldots\} \\
= \ & cum\{\sum_{k=1}^{K} s_k(t)e^{j\varphi_{mk}}, \sum_{k=1}^{K} s_k(t)e^{j\varphi_{nk}}, \sum_{k=1}^{K} s_k(t)e^{j\varphi_{pk}}, \ldots\} + cum\{n_m(t), n_n(t), n_p(t), \ldots\} \\
= \ & cum\{\sum_{k=1}^{K} s_k(t)e^{j\varphi_{mk}}, \sum_{k=1}^{K} s_k(t)e^{j\varphi_{nk}}, \sum_{k=1}^{K} s_k(t)e^{j\varphi_{pk}}, \ldots\}.
\end{aligned} \tag{13}
$$

Consequently, in order to simplify the illustration of the proposed method, we ignore the Gaussian noise in the following equations, and thus we can focus on the source signal.

Firstly let

$$\mathbf{x}(t) = \mathbf{J}\mathbf{y}^*(t), \tag{14}$$

where **J** is the exchange matrix with the size of $(2M+1) \times (2M+1)$ [21].

$$
\mathbf{J} = \begin{bmatrix}
0 & 0 & \ldots & 0 & 1 \\
0 & 0 & \ldots & 1 & 0 \\
\vdots & & & & \\
0 & 1 & \ldots & 0 & 0 \\
1 & 0 & \ldots & 0 & 0
\end{bmatrix} \tag{15}
$$

Under the assumptions that are made in the signal model part, we can calculate the fourth-order cumulant of the processed signal [8,14,16,17,22]:

$$
\begin{aligned}
& cum\{x_m(t), x_n^*(t), x_p(t), x_q^*(t)\} \\
= \ & E[x_m(t)x_n^*(t)x_p(t)x_q^*(t)] \\
& -E[x_m(t)x_n^*(t)]E[x_p(t)x_q^*(t)] \\
& -E[x_m(t)x_p(t)]E[x_n^*(t)x_q^*(t)] \\
& -E[x_m(t)x_q^*(t)]E[x_n^*(t)x_p(t)] \\
= \ & \sum_{k=1}^{K} c_{4s_k} e^{j[(m-n+p-q)\omega_k + (m^2-n^2+p^2-q^2)\phi_k]},
\end{aligned} \tag{16}
$$

$$
\begin{aligned}
& cum\{s_m(t), s_n^*(t), s_p(t), s_q^*(t)\} \\
= \ & \begin{cases} c_{4s_k}, & (m=n=p=q=k) \\ \\ 0, & (others) \end{cases}
\end{aligned}
$$

where $c_{4s_k}$ is the fourth-order cumulant of the source signal $s_k$.

We then can construct a cumulant matrix with the entries defined as follows:

$$
\begin{aligned}
\mathbf{C}(\bar{n}, \bar{m}) &= cum\{x_n(t), x_0^*(t), x_{-m}(t), x_m^*(t)\} \\
&= \sum_{k=1}^{K} c_{4s_k} e^{-j2\omega_k m} e^{j(\omega_k n + \phi_k n^2)},
\end{aligned} \tag{17}
$$

where

$$
\bar{m} = M + m + 1, \tag{18}
$$

$$
\bar{n} = M + n + 1, \tag{19}
$$

$$
m, n \in [-M, M] \tag{20}
$$

The cumulant matrix $\mathbf{C}$ can be written in the matrix form as:

$$
\mathbf{C} = \mathbf{A}_1(\boldsymbol{\theta}, \mathbf{r}) \mathbf{C}_{4s} \mathbf{A}_2^H(\boldsymbol{\theta}). \tag{21}
$$

$\mathbf{C}_{4s}$ is a diagonal matrix, and the diagonal elements are $c_{4s_1}, c_{4s_2}, \ldots, c_{4s_K}$ respectively:

$$
\mathbf{C}_{4s} = \begin{bmatrix} c_{4s_1} & 0 & 0 & \ldots & 0 \\ 0 & c_{4s_2} & 0 & \ldots & 0 \\ \vdots & & & & \\ 0 & 0 & 0 & \ldots & c_{4s_K} \end{bmatrix} \tag{22}
$$

$\mathbf{a}_1(\theta_k, r_k)$ $(\mathbf{a}_2(\theta_k))$ is the $k$th column of $\mathbf{A}_1(\boldsymbol{\theta}, \mathbf{r})$ $(\mathbf{A}_2(\boldsymbol{\theta}))$. They are $(2M+1) \times 1$ vectors given by

$$
\begin{aligned}
\mathbf{a}_1(\theta_k, r_k) &= [e^{j[(-M)\omega_k + (-M)^2 \phi_k]}, \ldots, \\
& \quad e^{j(M\omega_k + M^2 \phi_k)}]^T.
\end{aligned} \tag{23}
$$

and

$$
\mathbf{a}_2(\theta_k) = [e^{j2(-M)\omega_k}, \ldots, e^{j2M\omega_k}]^T \tag{24}
$$

Apply the EVD to the cumulant matrix $\mathbf{C}$, and we can have

$$
\mathbf{CU} = \mathbf{U\Sigma}, \tag{25}
$$

where

$$
\mathbf{\Sigma} = \begin{bmatrix} \lambda_1 & 0 & 0 & \ldots & 0 \\ 0 & \lambda_2 & 0 & \ldots & 0 \\ \vdots & & & & \\ 0 & 0 & 0 & \ldots & \lambda_{2M+1} \end{bmatrix}, \tag{26}
$$

$$
\mathbf{U} = [\mathbf{u}_1, \mathbf{u}_2, \ldots, \mathbf{u}_{2M+1}], \tag{27}
$$

and $\mathbf{u}_k$ is the $k$th corresponding eigenvector. Substituting Equation (21) into Equation (25), it can be calculated that

$$
\mathbf{A}_1(\boldsymbol{\theta}, \mathbf{r}) \mathbf{C}_{4s} \mathbf{A}_2^H(\boldsymbol{\theta}) \mathbf{U} = \mathbf{U\Sigma}. \tag{28}
$$

The rank of $\mathbf{C}$ is $K_1 + K_2 = K$, and there are only $K$ non-zero eigenvalues, which means that

$$
\lambda_{K+1} = \lambda_{K+2} = \lambda_{K+3} = \ldots = \lambda_{2M+1} = 0. \tag{29}
$$

Therefore, we have

$$\mathbf{A}_1(\boldsymbol{\theta}, \mathbf{r})\mathbf{C}_{4s}\mathbf{A}_2^H(\boldsymbol{\theta})\mathbf{u}_{K+1} = \mathbf{A}_1(\boldsymbol{\theta}, \mathbf{r})\mathbf{C}_{4s}\mathbf{A}_2^H(\boldsymbol{\theta})\mathbf{u}_{K+2} = \ldots = \mathbf{A}_1(\boldsymbol{\theta}, \mathbf{r})\mathbf{C}_{4s}\mathbf{A}_2^H(\boldsymbol{\theta})\mathbf{u}_{2M+1} = \mathbf{0}_{2M+1}. \quad (30)$$

We know that the ranks of $\mathbf{A}_1(\boldsymbol{\theta}, \mathbf{r})$ and $\mathbf{C}_{4s}$ are all $K$, which means that they are both full column-rank. Then it can be derived that

$$\mathbf{A}_2^H(\boldsymbol{\theta})\mathbf{u}_{K+1} = \mathbf{A}_2^H(\boldsymbol{\theta})\mathbf{u}_{K+2} = \ldots = \mathbf{A}_2^H(\boldsymbol{\theta})\mathbf{u}_{2M+1} = \mathbf{0}_{2M+1}. \quad (31)$$

Construct a matrix:

$$\mathbf{U}_{N1} = [\mathbf{u}_{K+1}, \mathbf{u}_{K+2}, \ldots, \mathbf{u}_{2M+1}]. \quad (32)$$

$\mathbf{A}_2^H(\boldsymbol{\theta})$ is orthogonal with $\mathbf{U}_{N1}$. Then the DOAs of all the sources (including both far-field and near-field) can be estimated with the following MUSIC spectrum:

$$\hat{\theta}_k = \arg \max_{\theta} \frac{1}{\|\mathbf{U}_{N1}^H \mathbf{a}_2(\theta_k)\|^2}. \quad (33)$$

In order to reduce the computational complexity of the algorithm, we here propose to apply the root-MUSIC for the estimation of the DOAs, which can avoid the spectrum search. Define

$$z = e^{j2\omega_k}. \quad (34)$$

Then Equation (24) can be rewritten as

$$\mathbf{a}_2(z) = [z^{-M}, \ldots, z^{M-1}, z^M]^T \quad (35)$$

Define a polynomial:

$$f(z) = z^{2M+1}\mathbf{a}_2^T(z^{-1})\mathbf{U}_{N1}\mathbf{U}_{N1}^H\mathbf{a}_2(z). \quad (36)$$

There is a conjugate symmetry property in this polynomial. The roots of this polynomial come in pairs which are inside or on the unit circle, and one root is the conjugate reciprocal of the other. Only one root of each pair will be selected. By ordering the roots inside the unit circle and taking the $K$ roots that are closest to and inside the unit circle, all the DOAs can be estimated.

*3.2. Range Estimation for Near-Field Sources*

Consider the definition of the eigenvector

$$\mathbf{A}_1(\boldsymbol{\theta}, \mathbf{r})\mathbf{C}_{4s}\mathbf{A}_2^H(\boldsymbol{\theta})\mathbf{u}_k = \lambda_k \mathbf{u}_k. \quad (37)$$

When the eigenvalue $\lambda_k$ is non-zero, the eigenvector $\mathbf{u}_k$ is the linear combination of all the columns of the steering matrix $\mathbf{A}_1(\boldsymbol{\theta}, \mathbf{r})$. Define a matrix

$$\mathbf{U}_S = [\mathbf{u}_1, \mathbf{u}_2, \ldots, \mathbf{u}_K]. \quad (38)$$

Obviously, $\mathbf{A}_1(\boldsymbol{\theta}, \mathbf{r})$ and $\mathbf{U}_S$ span the same column subspace. However, the ranges of near-field sources cannot be estimated directly through $\mathbf{U}_S$. Apply the Gram–Schmidt orthogonalization to $\mathbf{U}_S$:

$$\mathbf{u}_{k\perp} = \begin{cases} \mathbf{u}_1, & (k=1) \\ \mathbf{u}_k - \sum_{i=1}^{k-1} \frac{<\mathbf{u}_k, \mathbf{u}_i>}{<\mathbf{u}_i, \mathbf{u}_i>}\mathbf{u}_i, & (k>1) \end{cases}$$

With the orthogonalized eigenvectors, we can form another matrix whose columns are orthogonal with each other:

$$\mathbf{U}_{S\perp} = [\mathbf{u}_{1\perp}, \mathbf{u}_{2\perp}, \ldots, \mathbf{u}_{K\perp}]. \tag{39}$$

Assume that another matrix $\mathbf{U}_{N\perp}$ is orthogonal with $\mathbf{A}_1(\boldsymbol{\theta}, \mathbf{r})$, we can derive that

$$[\mathbf{U}_{S\perp}, \mathbf{U}_{N\perp}][\mathbf{U}_{S\perp}, \mathbf{U}_{N\perp}]^H = \mathbf{I}, \tag{40}$$

and

$$
\begin{aligned}
[\mathbf{U}_{S\perp}, \mathbf{U}_{N\perp}][\mathbf{U}_{S\perp}, \mathbf{U}_{N\perp}]^H &= \mathbf{U}_{S\perp}\mathbf{U}_{S\perp}^H + \mathbf{U}_{N\perp}\mathbf{U}_{N\perp}^H + \mathbf{U}_{S\perp}\mathbf{U}_{N\perp}^H + \mathbf{U}_{N\perp}\mathbf{U}_{S\perp}^H \\
&= \mathbf{U}_{S\perp}\mathbf{U}_{S\perp}^H + \mathbf{U}_{N\perp}\mathbf{U}_{N\perp}^H.
\end{aligned} \tag{41}
$$

Then we know

$$\mathbf{U}_{N\perp}\mathbf{U}_{N\perp}^H = \mathbf{I} - \mathbf{U}_{S\perp}\mathbf{U}_{S\perp}^H. \tag{42}$$

Substitute every DOA estimated before, and the corresponding range of the source can be estimated through the following MUSIC spectrum:

$$
\begin{aligned}
\hat{r}_k &= \arg\max_r \frac{1}{\mathbf{a}_1^H(\hat{\theta}_k, r_k)\mathbf{U}_{N\perp}\mathbf{U}_{N\perp}^H\mathbf{a}_1(\hat{\theta}_k, r_k)} \\
&= \arg\max_r \frac{1}{\mathbf{a}_1^H(\hat{\theta}_k, r_k)(\mathbf{I} - \mathbf{U}_{S\perp}\mathbf{U}_{S\perp}^H)\mathbf{a}_1(\hat{\theta}_k, r_k)}.
\end{aligned} \tag{43}
$$

For the substituted DOA $\hat{\theta}_k$, if the source lies in the Fresnel region, the corresponding estimate of the range can be obtained. But if the substituted DOA $\hat{\theta}_k$ is for the far-field source, the estimated range $\hat{r}_k$ is out of the Fresnel region. With this standard, the far-field and near-field sources can be distinguished easily. For the near-field sources, the estimated DOA and range are automatically paired, requiring no extra pairing algorithms.

The proposed method can be summarized as follows:

Step 1: Construct the cumulant matrix $\mathbf{C}$.
Step 2: Apply the EVD to $\mathbf{C}$.
Step 3: Obtain $\mathbf{U}_{N1}$ orthogonal to $\mathbf{A}_2(\boldsymbol{\theta})$.
Step 4: Estimate the DOAs $\hat{\theta}_k$ ($k = 1, 2 \ldots, K$).
Step 5: Apply the Gram-Schmidt orthogonalization to $\mathbf{U}_S$.
Step 6: By substituting the $k$th estimated DOA, estimate the $k$th range.
Step 7: Repeat Step 5 until all the $K_2$ range estimates are all obtained.

*3.3. Complexity Analyses*

In this part, we will analyze the computational complexity, and make a comparison with other methods. The main different computational complexities of the methods are compared in the following Table 1:

**Table 1.** Main complexities of different methods.

| Item | Proposed Method | MOS | HOS |
|---|:---:|:---:|:---:|
| Matrix construction | 1 | 2 | 2 |
| EVD | 1 | 2 | 2 |
| Gram-Schmidt orthogonalization | 1 | 0 | 0 |

For MOS, a fourth-order cumulant matrix is designed with the size of $[(\frac{M}{2} + 1)^2 + 1] \times [(\frac{M}{2} + 1)^2 + 1]$, whose computational complexity is about $9((\frac{M}{2} + 1)^2 + 1)^2 T$. Then a covariance matrix with the size of $(2M + 1) \times (2M + 1)$ is constructed, and the complexity is about $(2M + 1)^2 T$. Then the eigenvalue

decomposition is applied to the two matrices respectively (the complexities are about $\frac{4}{3}((\frac{M}{2}+1)^2+1)^3$ and $\frac{4}{3}(2M+1)^3$). Therefore, the whole computational complexity of MOS is about $O(9((\frac{M}{2}+1)^2+1)^2T+(2M+1)^2T+\frac{4}{3}((\frac{M}{2}+1)^2+1)^3+\frac{4}{3}(2M+1)^3)$. Similarly for HOS, two fourth-order cumulant matrices are constructed with the size of $(2M+1)\times(2M+1)$, and the eigenvalue decomposition is applied twice. The corresponding complexity is about $O(18(2M+1)^2T+\frac{8}{3}(2M+1)^3)$. The proposed method constructs only one $(2M+1)\times(2M+1)$ cumulant matrix, and applies the EVD once. But it also requires the application of Gram–Schmidt orthogonalization (the complexity is about $\frac{1}{2}(K-1)KM$), leading to the complexity about $O(9(2M+1)^2T+\frac{4}{3}(2M+1)^3+\frac{1}{2}(K-1)KM)$.

## 4. Simulation Results and Analysis

First of all, the computational efficiency of the different methods is studied. A laptop was adopted as the platform, with the CPU being an i7 (2.3 GHz) and RAM 8 GB. The simulation was carried out in the situation where there were one far-field sources and two near-field ones. A total of 500 snapshots were received with the 5-sensors-array, and 200 simulations were carried out. The average processing time of the methods are shown in Table 2. It can be observed that the efficiency of the proposed method is better than HOS and MOS.

**Table 2.** Average processing time (seconds) of different methods.

| Methods | Proposed Method | HOS | MOS |
|---------|-----------------|--------|--------|
| Time (s) | 0.1421 | 0.1711 | 0.2338 |

Secondly, the performance of the proposed method is examined. The results will be compared with other existing methods to show the effectiveness of our proposed method. The relationship between the root mean square error (RMSE) of estimation and signal–noise ratio (SNR) is adopted in the paper to examine the performance. RMSE is defined as follows:

$$RMSE = \sqrt{\frac{\sum_{p=1}^{P}|\hat{\alpha}_p - \alpha_{true}|^2}{P}}, \tag{44}$$

where $\hat{\alpha}_p$ is the estimation result of the $p$th trial, $\alpha_{true}$ is the true value, and $P$ is the number of independent Monte Carlo trials. The definition of SNR is given by

$$SNR = 10\log_{10}\frac{\sum_k^K s_k^2}{\varepsilon^2}, \tag{45}$$

where $\varepsilon^2$ is the noise variance and $s_k^2$ the power of the $k$th signal.

Consider the situation where four sources (two far-field and two near-field) are set at $[5°]$, $[25°]$, $[-10°, 1.5\lambda]$, and $[40°, 2\lambda]$. The ULA used in the simulation is made of 9 elements and the value of $d$ is selected as $\frac{\lambda}{4}$. The ULA receives 200 snapshots from the sources. The SNR varies from 0 to 30 dB, and the results with 200 independent Monte Carlo trials are shown in Figures 2–4. Besides, the performance of the proposed method is also compared with the Cramer–Rao bound (CRB) given in [23].

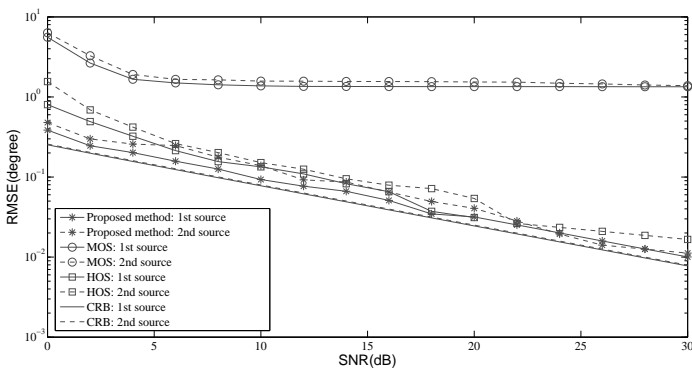

**Figure 2.** RMSE versus SNR: DOA of far-field source.

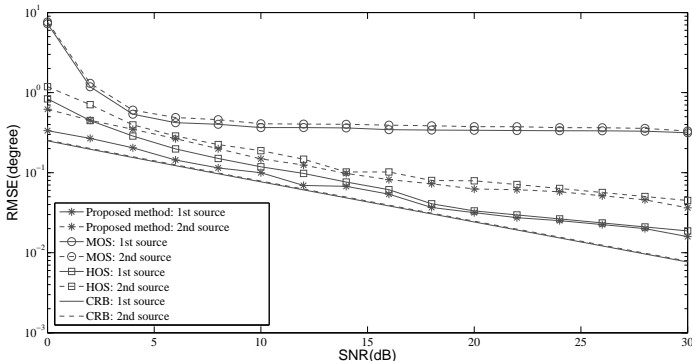

**Figure 3.** RMSE versus SNR: DOA of near-field source.

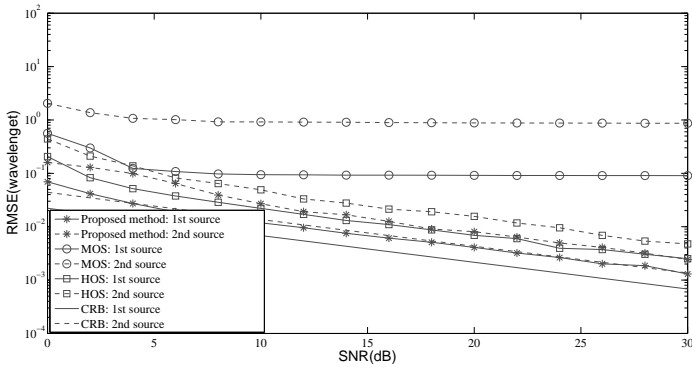

**Figure 4.** RMSE versus SNR: range of near-field source.

In Figures 2 and 3, it can be seen that the RMSEs of DOA estimation decreases when the SNR grows. The accuracy is very excellent, and is almost the same as the CRB due to the resistance of high-order cumulant. When the SNR is high enough, we can see that the difference between the CRB and the RMSE of the proposed method can be ignored. It can also be seen that the estimation of DOAs of different sources are nearly the same.

For different methods, it is obvious that MOS does not perform as well as other methods. HOS outperforms MOS, and the performance is very similar to the proposed method, but HOS leads to a much higher complexity than the proposed method. In Figure 4, the performance of the range estimation is displayed. The comparison is very similar to that of DOAs. The proposed method provides the best accuracy.

## 5. Conclusions

In this paper, we proposed a modified 2D MUSIC for mixed-field source localization based on a high-order cumulant. Different from other MUSIC-based methods where several Hermitian matrices are constructed, only one non-Hermitian cumulant matrix is required and one EVD is applied to estimate the DOAs and ranges of the mixed-field sources. The analysis shows that the proposed method results in a lower computational complexity than other existing methods, while the simulation results show that the proposed method can effectively achieve the decoupled estimation of the DOA and range.

**Author Contributions:** The work presented here was carried out in collaboration between all authors. The general idea was proposed by M.M.. Z.S. performed the simulations and analyzed the results. M.M. wrote the paper. Z.S. revised the manuscript and provided many valuable suggestions. All authors have read and agreed to the published version of the manuscript.

**Acknowledgments:** This work was partially supported by the National Natural Science Foundation of China through grants (61471021), and partially supported by the National Defense Basic Research Program (JCKY2017603B006).

**Conflicts of Interest:** The authors declare no conflict of interest.

## Abbreviations

The following abbreviations are used in this manuscript:

| | |
|---|---|
| DOA | Direction of Arrival |
| 2D | Two-Dimension |
| MUSIC | MUltiple SIgnal Classification |
| ESPRIT | Estimation of Signal Parameters via Rotation Invariant Technique |
| EVD | EigenValue Decomposition |
| SVD | Singular Value Decomposition |
| ULA | Uniform Linear Array |
| SNR | Signal-to-Noise Ratio |
| RMSE | Root Mean Square Error |
| CRB | Cramer-Rao Bound |

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
