# Peer review of "Mixed-Field Source Localization Based on the Non-Hermitian Matrix"

_information, doi:10.3390/info11010056_

Round 1

Reviewer 1 Report

In this paper, a high-order cumulant MUSIC-like method is proposed  for mixed-field source localization. This method is based on a non-Hermitian matrix et estimates the DOA and range in a decoupled way. The mathematical development is straightforward, there are some small novelties in the paper. However, there are many problems in the paper

The paper has strong similarities with the following paper:

Li, J.; Wang, Y.; Le Bastard, C.; Wei, G; Ma, B.; Sun, M.; Yu, Z. Simplified High-order DOA and Range Estimation with Linear Antenna Array. IEEE Commun. Lett. 2017, 21, 76–79.

The authors have to added this reference in the revised paper, and specify what are the main difference of the proposed method with the method of the reference.

The English writing is very poor, several examples (there are too much errors in the paper):

‘eivengvectors’ have been used many times, however, it should be eigenvectors. Line 117, ‘due to the resistance of high-order cumulant’ should be ‘due to the resistance to Gaussian noise of  high-order cumulant’. In line 120, ‘it can be studied that the estimation of DOAs of different sources are nearly the same’ should be ‘it can be observed that the performance of the estimation of DOAs of different sources is nearly the same’. In line 131, ‘Hermitian matrices are constructed, Only one non-Hermitian’, should be ‘Hermitian matrices are constructed, only one non-Hermitian’. ….

The mathematical development is not rigorous and has several serious problems

In Eq.7, the dimension of vector y is 2M+1, and in Eq13, this dimension becomes to 2M, which is not consistant. Why introduce vector x? You can use vector y in the whole paper with the same conclusion! In Eq(25), you define U until u_K (K eigenvectors associated with the K largest eigenvalues). However, in Eqs. (28) and (29) you use u_(K+1), u_(K+2) -à the mathematical development and writing are not rigorous. Eq(29) should be rewritten correctly. Eq(33) is totally wrong, please read a good paper concerning root_MUSIC for understanding why.

Author Response

Dear Reviewer,

  Thank you very much for your review and advice. We have cited the paper as Ref [17] and specify the difference. We are sorry about the writing mistakes, and have tried our best to correct them. And we have read some paper about root-MUSIC, and correct the equations. Please check the figures below, as well as the revised paper.

Best regards,

Minggang Mo

Reviewer 2 Report

In this paper, a music-like approach for mixed-fixed source localization is studied. The authors stated that their approach has lower computational complexity than that of others studied in the literature. However, the details of where the exact complexity comes have been largely missing in the paper. The authors also conducted experimental work to show the efficiency of their approach. 

Some concrete comments:

L. 2: "based high-order"=>"based on high-order"

throughout the paper: "eivengvector"=>"eigenvector", "can not"=>"cannot"

L. 32: "to construct"=>"constructing"

L. 42: "Part"=>"Section"

L. 64: "independent from"=>"independent of"

In Equality (29): "==..."=>"=...="

Ls. 103-108: Could you give more details on where the complexity comes? 

L. 131: "Only"=>"only" 

Author Response

Dear Reviewer,

  Thank you very much for your review and advice. We have corrected the mistakes in the paper. And we have also added the details of the complexity and a simulation to show the efficiency of the proposed method. Please check the figures below, as well as the revised paper.

Best regards,

Minggang Mo

Reviewer 3 Report

The authors propose a root-MUSIC algorithm for mixed-field (near- and far-field) source localization. The work is based on existing work on mixed-field MUSIC, and the main novelty lies in the application of root-MUSIC to the problem.

Major comments:

The most crucial step in the algorithm, i.e. how to form the polynomial in (33) is insufficently explained. The evaluation is a very limited, theoretical and only a small proof-of-concept. Also, showing results for more than only one near- and far-field source would add value.
Minor comments:
The step in (13) omitting the noise is not completely understandable, and should be explained and motivated more carefully. above (12): reference missing

Author Response

Dear Reviewer,

  Thank you very much for your review and advice. We have added some explanation for the polynomial and the reason why the noise can be omitted. And we also run the simulation again with 4 sources (2 far-field and 2 near-field) to examine the RMSE. Please check the figures below, as well as the revised paper.

Best regards,

Minggang Mo

Round 2

Reviewer 1 Report

The authors have made a big effort to improve the writing of the paper. However, the equation of root-MUSIC is still wrong. In fact, Eq(36) is not a polynomial, the true equation can be found in many classical books or articles (a2(1/z)T Pi_n a2(z)). In addition, there are still too many English errors, for example (only for example, there are really too much writing problems)

In line 28, ‘construct several two cumulant matrices’ should be ‘construct two cumulant matrices’ In line 46, ‘Several simulation is carried out’ should be ‘Several simulations are carried out’ Line 71, ‘fourth-order cumulatn’ should be ‘fourth-order cumulant’ Line80, ‘kth columns A1(q, r)’ should be ‘kth column of A1(q, r)’

Author Response

Dear Reviewer,

  Thank you very much for your review and advice. We checked again the paper and correct some more writing mistakes, especially in the introduction part. We have also rewritten the polynomial as follows, which is the modification of the original one.

Best regards,

Minggang Mo

Reviewer 2 Report

The authors fixed the problems pointed out in my comments. I am satisfied with the current cersion and it should be accepted for publication.

Author Response

Dear Reviewer,

  Thank you very much for your review and positive comments.

Reviewer 3 Report

The reviewers addressed the major issues accordingly. 

I have a few additional comments:

Please explain why the signal vector needs to be flipped and conjugated with (14). Does this have any effect? Please define the indices m,n,p in (12). I assume they correspond to microphone indices?

Author Response

Dear Reviewer,

  Thank you very much for your review and advice. We have added the definition of Equation (12).

  The signal is flipped and can resist the noise better according to Reference [21] when it is added with the cumulant matrix constructed with the original signal. In our paper, we want to verify this pre-processing (flipping) can work properly with our method, and we are going to do a deeper research of this noise-resisting technique in the future.

Best regards,

Minggang Mo

Round 3

Reviewer 1 Report

The authors have given a satisafactory response to all my questions, I recommande the acceptatio of the paper.

One point: Eq(36) for root-MUSIC is still wrong, it should be

a2^T (z^-1) (UnHn^H)a2(z), it is written in every classical article. The authors should make the necessary corrections!

Author Response

Dear Reviewer,

  Thank you very much for your careful review. We are so sorry for the careless mistake. We have corrected Equation (36) and checked again the paper.

Best regards,

Minggang Mo
